# Operating and Dynamic Capabilities and Their Impact on Operating and Business Performance

Jasna Prester 

Faculty of Business and Economics, University of Zagreb, 10000 Zagreb, Croatia; jprester@efzg.hr

**Abstract:** This work verified, through confirmatory factor analysis, a new measurement model for measuring dynamic capabilities based on current propositions in the literature, using a database of 1008 manufacturing sites from 16 countries. The indirect and direct effects of dynamic capabilities on ordinary capabilities and operating and business performance were also checked. In particular, we tested whether there were any mediating or moderating effects between ordinary and dynamic capabilities on operating and business performance. All the tests were performed through SEM in AMOS and OLS in SPSS. Additionally, a Heckman two-step procedure was performed. The proposed measurement model shows a good fit, meaning that it can be used for further exploring the interplay of ordinary and dynamic capabilities. The mediating and moderating effects of dynamic capabilities measured showed only partial mediation and only low and nonsignificant levels of moderation, meaning that further analysis of their interrelationships on performance should be investigated. Measurement models for dynamic capabilities are especially scarce. Virtually no work deals with dynamic capabilities in the field of operations management; yet it is exactly by means of operations that one can verify the dynamic capabilities being used and what benefits they bring.

**Keywords:** ordinary capabilities; dynamic capabilities; operations performance; business performance; GMRG V

## 1. Introduction

Even the most up-to-date research [1] shows that despite constant research in the field of strategizing manufacturing companies, there is still no prescription for how to achieve sustainable competitive advantage and how one company can become more competitive than its rivals. More than two decades ago, Refs. [2,3] introduced the notion of dynamic capabilities (DCs) that could produce a competitive advantage. In 2015, Pisano [4], also a great contributor to dynamic capability theory, argued that capabilities are still not adequately described and researched and that there is still no unique definition of what constitutes dynamic capabilities. His analysis showed that most authors try to define dynamic capabilities (which, in his view, is also important), but argues that there is no work to describe how to build these capabilities. Since it is known that only what is measured can be improved, our first aim is to propose a measurement model for dynamic capabilities. As Pisano [4] states, there is more literature on what should constitute dynamic capabilities than proposed measurement models. So far, we have found only three empirical papers [5–7] and one theoretical proposition [8]. More recently, we found [9,10] to measure dynamic capabilities. Moreover, refs. [11,12] argue that dynamic capabilities do not, per se, create competitive advantages; rather, dynamic capabilities build on operating capabilities to create competitive advantages. Ordinary and operating capabilities are used interchangeably by [13]. Therefore, there is a clear gap in the literature as to how one may measure dynamic capabilities, how they affect operating capabilities, and how they affect firm performance [10,14]. In this work, we propose and test a measurement model of dynamic capabilities that can serve as a basis for improving the dynamic capabilities of a manufacturing company.

A dominant stream of the literature exploring dynamic capabilities as a source of competitiveness is in the domain of strategic management. But, as [15] points out, it is the "details" of how these capabilities are used that create competitive advantages. These "details", as [15] calls them, and where the impact of DCs can be seen, lie in their influence on routines, practices, and operating capabilities. This is exactly what the operations management literature deals with. Our contribution is, to our knowledge, the first to relate DCs to operating or "ordinary" capabilities dealt with in the operations management literature. There is a discrepancy in the terminology of what is a dynamic capability, as can be seen from the tables provided in [16–18]. All these consulted papers are theoretical and do not propose measurement models. Our primary goal, and a significant contribution of our study, is to propose a measurement model for discussion and further refinement to better understand ordinary and dynamic capabilities and their mutual relationships, as well as their effect on firm performance. To cite [14], dynamic capabilities are essential, but it is how they are deployed on operating capabilities or practices that will drive a company toward a more competitive position. The value is in the derived measurement model from the recent literature research on dynamic capabilities. So, our first research questions and contributions are as follows:

RQ1—How may one measure dynamic capabilities, and how do they affect performance?

So far, we know that DCs indirectly affect operating capabilities, but there is still no agreement on how these dynamic capabilities affect operating capabilities [14–16,19]. To put it in statistical terms, one must research the DCs' mediating or moderating effects on operating capabilities in relation to financial and business performance or whether DCs even have quadratic effects, as [20] proposes. Therefore, we pose a second research question:

RQ2—What is the nature of the relationship between operating capabilities and dynamic capabilities on operations and business performance?

Most research was conducted in the high-tech sector, and most of this research explored how to enhance innovation through the use of dynamic capabilities. There are no insights from "ordinary" manufacturing sectors that are not in such high-technology-intensity fields. Our results explain the role of capabilities and dynamic capabilities in affecting performance, in a general sense, across many manufacturing sectors and economy types. According to [4], most of the manufacturing is represented by relatively stable competition between a few known rivals who compete in relatively well-defined markets.

Our model analyses how capabilities affect operations and business performance. The Global Manufacturing Research Group (GMRG) V is used. This large GMRG V collection round resulted in a database of 1008 manufacturing companies from 16 countries. All results are generated by means of structural equation modeling in AMOS and with OLS in SPSS to check the consistency of results. Additionally, we checked selectivity bias with a Heckman two-step regression model and obtained consistent results.

Standard tests were performed, and a good model fit was obtained. Thus, this paper contributes to the empirical stream of research trying to develop a measurement model of dynamic capabilities. Once the measurement model is tested, concrete managerial implications can be provided. This paper might be interesting to academia for further refinement of the model and to practitioners in terms of how to enhance their capabilities.

The research synthesis can be presented in the following way, described by the structure of this paper:

(1) Perform a literature search for all articles in WoS, Scopus, and Google Scholar that have "dynamic capabilities" in their title;
(2) Only extract papers that potentially deal with the measurement instrument;
(3) Compile a measurement instrument from found works;
(4) Compare each question to the GMRG V instrument;
(5) Perform a confirmatory factor analysis of the instrument;
(6) Test if the operating and dynamic capabilities improve performance, because the literature states that they should;

(7)  Test the mediating and moderating effects of dynamic capabilities on operating capabilities, because the literature is not clear on this;

(8)  Provide recommendations.

This paper is organized as follows: first, we present and examine the current literature on operations capabilities measurement and dynamic capabilities. From this literature, we construct a table that shows the overlapping of concepts by use of the GMRG research instrument. Then, we propose a measurement instrument that is organized as operating and dynamic capabilities and how they affect operating and business performance on the other side. The results include testing the hypothesized base model as well as the verification of the mediating and moderating effects of dynamic capabilities on performance. A discussion, managerial implications, and the conclusion follow.

## 2. Related Literature

The literature on operating and dynamic capabilities is scarce [21] and is focused more on innovation. In an attempt to describe capabilities, we try to review studies that will enable us to propose a measurement model and thus research hypotheses. While both topics (operating and dynamic capabilities) are mentioned in the literature quite often, measurement models are scarce, so in the following section, we specifically select those that propose measurement models.

### 2.1. Ordinary Capabilities

Ordinary or operations capabilities [13] enable a company to perform its daily activities. Zollo and Winter [12] refer to these as routines, but in 2011, ref. [15] renamed them capabilities and defined them as the capacity to perform a particular task in a reliable and satisfactory manner. Operations capabilities or first-order capabilities [22] are defined as capabilities that enable a manufacturing company to work in the present. Operations capabilities, according to [23], are well established in the operations management literature and are measured in terms of quality, cost, flexibility and delivery. Operations capabilities are not directly measurable [24]. A capability is an ability to undertake an activity. Only after the activity is performed can it be observed, and its results assessed. According to [25], it is absolutely necessary to clearly define constructs in order to study relationships. Nevertheless, this is also not an easy task, because there is still a lack of consensus about what constitutes dynamic and operating capabilities. For example, ref. [8] researched the gap concerning the measurement of operations and dynamic capabilities. Careful analysis of their work revealed that in their operations capability constructs, there are also dynamic capabilities [26], as well as intellectual capital [27].

According to [26,28], operations capabilities are rooted in (1) the skills of personnel, (2) facilities and equipment, (3) processes and routines including technical manuals, and (4) the administrative coordination needed to get the job done [29]. Operations capabilities will be considered high if a company has (1) a skilled workforce, (2) state-of-the-art equipment, (3) clearly defined and available descriptions of activities to all employees and (4) state-of-the-art coordination. These operating capabilities are based on competitive priorities in terms of quality, speed, flexibility, and cost. Each one can be measured. For example, the scrap rate, efficiency through costs, volume or variety of products can be monitored daily. By themselves, they do not present a competitive advantage, as many market players can achieve this same quality and better.

Wang et al. [30] give examples of quality control and continuous improvement in a company. Quality control is a daily control activity necessary for at least maintaining the quality at some level or controlling for problems in quality. Implementing continuous improvement, lean management or TQM in a company will affect every employee and their engagement. Little by little, these initiatives will reduce the scrap rate, waste, and unnecessary expenses, and therefore improve the company's performance. Since all of these improvements are applied to operating capabilities, little by little, the company acquires a dynamic capability and becomes better than its competitors [31] Helfat and Winter [15]

use the example of Intel. This company's ability to constantly innovate [15] demonstrates that it has dynamic capability, built upon investments into R&D, education of their highly skilled workforce, and the development of routines for even faster innovation. However, ref. [26] warns that investment into R&D, education and state-of-the-art equipment represent operating capabilities, because a better-performing company can invest further, but the ability to constantly innovate is a dynamic capability because many companies have invested into R&D, employees and equipment but have still failed to innovate, meaning they did not manage to transform operating capabilities into dynamic capabilities.

*2.2. Dynamic Capabilities*

Zollo and Winter [12], in their work, draw on that of [32], in which operational and administrative routines, by aid of the deliberate updating of these routines, form dynamic capabilities. There is a discrepancy in the terminology regarding what is a dynamic capability, as can be seen from tables provided by [15–18].

Verification of constructs and relationships among organizational capabilities lags far behind conceptual and theoretical developments [24,33]. Ellonen et al. [34] state that the role of dynamic capabilities is still unclear. So far, only [5–7,9,10] have proposed a measurement model for dynamic capabilities. This is why we chose to contribute to the empirical validation of dynamic capabilities and, on the grounds of our findings, to contribute with prescriptions for how to build dynamic capabilities in practice. Our work differs from [9,10], as they only proposed measurement models and tested only the direct link with performance.

Teece [26] warns that even dynamic capabilities can become ordinary (operating) capabilities when other companies manage to replicate them. He illustrates it with Toyota Production System (TPS). Toyota had a competitive advantage for decades but, as more and more competitors introduced lean management into their practices, their dynamic capability became ordinary capabilities. According to [6], dynamic capability does not directly increase business performance or competitive advantage. There are indirect effects, and these suggest that dynamic capabilities mediate operating capabilities and through this indirect effect, a company becomes more competitive. In other words, operating capabilities create competitive advantages through the help of dynamic capabilities. This is in line with [26,35], who also propose a mediating relationship. The present study contributed by testing these posited relationships. Recent proposed measurement models [9,10] only tested the direct relationship with performance. Therefore, we contribute to theory while also testing the indirect effects of dynamic capabilities on operating capabilities.

Teece [26] states [36] as the first author stating that dynamic capabilities are second-order capabilities. He defines dynamic capabilities as abilities related to sensing, seizing, and reconfiguring: (1) sensing means seeing opportunities in the market, (2) seizing involves mobilizing resources to address opportunities, and (3) reconfiguring involves reorganizing resources already in possession to perform those new tasks. That would mean that a company with highly dynamic capabilities will enable its employees to respond more quickly to changes.

Wang et al. [5], based on the work of [26], define dynamic capabilities as mixing external new knowledge with existing knowledge. Reconfiguration allows a firm to use newly acquired knowledge for the production of new products or for increasing the quality of existing processes in place. In [5], dynamic capabilities change in time, so this needs to be addressed additionally. To address path dependence and the question of time in building capabilities, we borrowed and used the methodology from [37]. They state that time-dependent research may be analyzed with cross-sectional data (p. 4886) with the argument that in the sample, there will be companies at all levels of capabilities.

### 2.3. Operations Performance

Narasimhan and Das [38–40] agree that operating capabilities should be measured through the standard operating priorities of cost, quality, delivery, flexibility and innovation in comparison to their competitors.

It must be noted that today, successful companies may engage in multiple performance objectives [41], in contradiction to the sand cone model [37], which claims that capabilities are built in a specified order.

Manufacturing performance is measured as a second-order construct including several questions addressing cost, quality, delivery, customization, and new product introduction time, in line with [42].

### 2.4. Business Performance

Dynamic capabilities are mostly researched in the strategic management literature [43], and a usual measure of competitiveness is business performance [44]. Therefore, the analysis would not be complete without some measurement of competitiveness through business performance [45]. We adopted the measurement scale from [6], which includes sales growth, profitability growth and market share growth. Responders had to rate their performance on each question from 1 (decreased more than 25% in the last two years) to 7 (increased more than 25% in the last two years).

## 3. Hypothesized Model

So far, we only know that operating and dynamic capabilities should have a positive relationship with operating performance and business performance, as hypothesized by [2] and [3]. A recent study shows that lean capabilities enhance performance if they are considered as dynamic capabilities [46], or if technology is considered as a dynamic capability it can enhance performance [47]. Meanwhile, a purely theoretical recent paper [48] also claims that dynamic capabilities enhance performance. The authors tried to propose a measurement model, but not in line with original work on dynamic capabilities, and also used relatively old references. Therefore, we first started by testing our measurement instrument, and then checked how operating and dynamic capabilities affect performance. No such study has yet been performed, since there is still not an accepted measurement model. Model 1 in Figure 1 could be such as model. However, there is still no agreement regarding how these dynamic capabilities affect operating capabilities [14–16,19]. There is a disagreement over whether they have a mediating or a moderating effect. To answer our second research question on how DCs affect operating capabilities, we constructed Model 2, which tests mediating effects. Model 3 tests moderating effects in order to clarify concurrent findings.

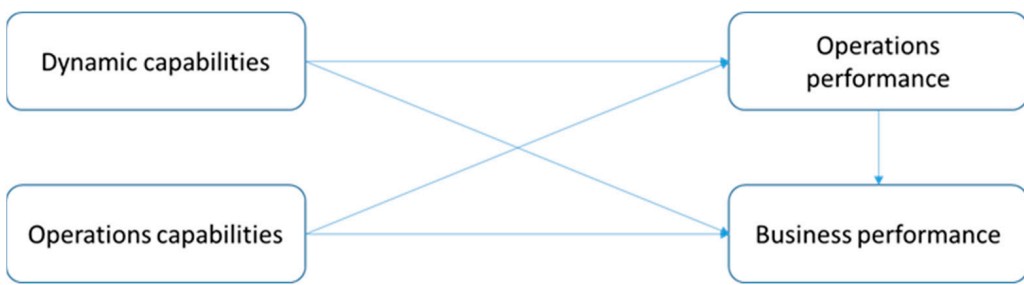

**Figure 1.** Testing model direct relationships.

Model 1 in Figure 1 hypothesizes the direct relationship between operating and dynamic capabilities and operating and business performance. This model enables us to check recent findings from [9,10] to measure dynamic capabilities, but tests only the direct relationship with performance. Grünbaum and Stenger [49] found no direct link between DCs and performance. However, the focus of their work was how DCs affect the innovation of small companies. Model 2 in Figure 2 and Model 3 in Figure 3 are relatively

scarcely researched. For example, ref. [15] calls this relationship "blurry", without explaining whether it is a mediating or moderating relationship, while [6] only proposes a mediating role. The authors of [14,16], in their theoretical papers, only hypothesize that the relationship is indirect, but do not mention whether it is a mediating or a moderating relationship. Model 2 builds on Model 1 and adds the mediating effect of DCs and operations capabilities. Pavlou and El Sawy [7] also proposed a mediating effect, but they concentrated on the mediating role of environmental dynamism on new product performance, not on the performance of the company to test competitiveness. On the other hand, refs. [50,51] found no mediating relationship, and instead found that DCs are context-dependent; ref. [51] found that organizational competencies are the moderating variable. Therefore, in Model 3, we check moderation in line with [31], which states that dynamic capabilities are higher-order constructs.

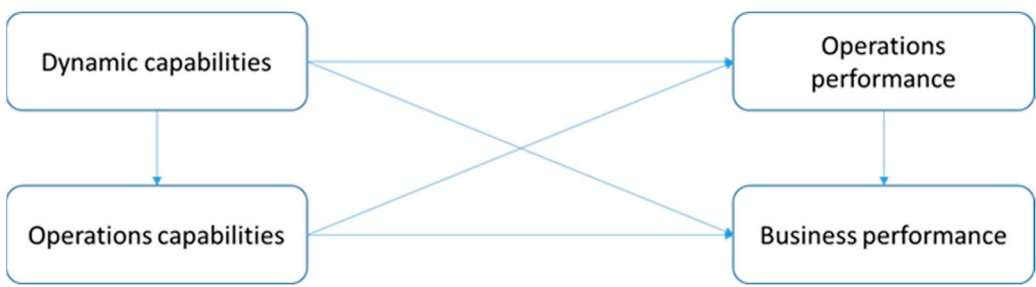

**Figure 2.** Testing Model 2—mediation.

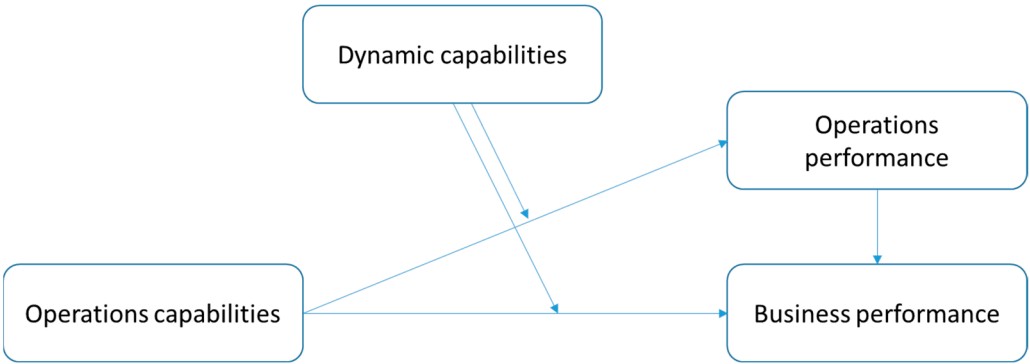

**Figure 3.** Testing Model 3—moderation.

All hypothesized models represent relatively new perspectives in dynamic capability empirical research, in particular regarding direct and indirect relationships. It is necessary to investigate direct versus indirect effects on performance. Since there is a large discrepancy in the literature, we cannot state which model should be the preferred model; rather, we must test all three models. Therefore, we have a set of three hypotheses.

**H1.** *Dynamic capabilities directly and positively affect operating and business performance.*

**H2.** *Dynamic capabilities mediate operations capabilities by positively affecting operating and business performance.*

**H3.** *Dynamic capabilities moderate operations capabilities by positively affecting operating and business performance.*

## 4. Methodology

### 4.1. Measurement Propositions of Dynamic and Ordinary Capabilities in the Literature

To measure operating capabilities, we adopted the framework presented by [38,39], who state that operating capabilities should be measured through standard operating priorities of cost, quality, delivery, flexibility, and innovation in comparison to competitors. The authors in [31,32,39] also all agree that operating capabilities should be measured through standard operating priorities of cost, quality, delivery, flexibility, and innovation in comparison to competitors. Therefore, we measured operating capability through standard operating priorities of cost, quality, delivery, flexibility, and innovation in comparison to competitors as a second-order construct.

There is clearly much research on dynamic capabilities (though mostly from only a theoretical perspective), but only a few studies propose measurement items. In Table 1, we summarize four research studies that proposed concrete, testable variables for measuring dynamic capabilities. Our first goal was to see how these constructs are similar, and how the GMRG survey can be used for investigating dynamic capabilities. To achieve this, we first constructed Table 1, and then we compared each question with the GMRG questionnaire. Then, in the fourth column of Table 1, we include the GMRG variable name if it exists in the GMRG questionnaire.

**Table 1.** Measurement of dynamic capabilities in the literature.

| Authors | Base Literature | Measurement Items | GMRG Survey Item | Researched Industry |
|---|---|---|---|---|
| Wang et al. (2015) [5] | Adapted from Pandza and Holt (2007) [52], Wang and Ahmed (2007) [30] and Garcia-Morales et al. (2008) [53]. | Absorptive capability | | |
| | | This firm has the necessary skills to implement newly acquired knowledge. | CG09.2 | |
| | | This firm has the competences to transform the newly acquired knowledge. | FC4.3 | |
| | | This firm has the competences to use the newly acquired knowledge. | CG09.4 | |
| | | Transformative capability | | |
| | | People in this firm are encouraged to challenge outmoded practices. | FC4.3 | 113 UK SME |
| | | This firm evolves rapidly in response to shifts in our business priorities. | FC2.1 | |
| | | This firm is creative in its methods of operation. | CG09.4 | |
| | | This firm seeks out new ways of doing things. | CG09.5 | |
| | | People in this firm get a lot of support from managers if they want to try new ways of doing things. | I13d | |
| | | This firm introduces improvements and innovations in our business. | I06.1-5 | |
| Protogerou et al. (2011) [6] | 7-point Likert scale: 1-not using to 7-used extensively | Coordination capability | | |
| | | Integration and standardization of business processes | I13.5-8 | |
| | | Adoption of the latest management tools and techniques | CG09.1 | |
| | | Systematic implementation of business plan | FC2.2 | |
| | | Learning capability | | 271 Greek manufacturing companies |
| | | Organized processes of in-house learning and knowledge development | I03 | |
| | | Systematic on the job training, efficient team working | I03 | |
| | | Strategic competitive response capability | | |
| | | Effective benchmarking | - | |
| | | Systematic formulation of long-term strategy | FC2.1 | |

**Table 1.** *Cont.*

| Authors | Base Literature | Measurement Items | GMRG Survey Item | Researched Industry |
|---|---|---|---|---|
| Pavlou and El Sawy (2011) [7] | 7-point Likert scale in comparison to competitors | Timely response to competitive strategic moves | FC8.4 | 180 New product business unit |
| | | Flexible adaptation of human resources to technological and competitive changes | I 13.9-12 | |
| | | Sensing capability | | |
| | | We frequently scan the environment to identify new business opportunities. | FC8.4 | |
| | | We periodically review the likely effect of changes in our business environment on customers. | FC8.4 | |
| | | We often review our product development efforts to ensure they are in line with what the customers want. | FC8.3 | |
| | | We devote a lot of time implementing ideas for new products and improving our existing products. | FC8.1-3 | |
| | | Learning capability | | |
| | | We have effective routines to identify, value, and import new information and knowledge. | I13.5-8 | |
| | | We have adequate routines to assimilate new information and knowledge. | FC2.1, | |
| | | We are effective in transforming existing information into new knowledge. | FC2.2. | |
| | | We are effective in utilizing knowledge into new products. | FC8.4. | |
| | | We are effective in developing new knowledge that has the potential to influence product development. | FC8.1-3 | |
| | | Integrating capability | | |
| | | We are forthcoming in contributing our individual input to the group. | I13.4 | |
| | | We have a global understanding of each other's tasks and responsibilities. | I13.2 | |
| | | We are fully aware who in the group has specialized skills and knowledge relevant to our work. | I13.3 | |
| | | We carefully interrelate our actions to each other to meet changing conditions. | FC2.1. | |
| | | Group members manage to successfully interconnect their activities. | FC2.2. | |
| | | Coordinating capability | | |
| | | We ensure that the output of our work is synchronized with the work of others. | I13.7 | |
| | | We ensure an appropriate allocation of resources (e.g., information, time, reports) within our group. | I13.8 | |
| | | Group members are assigned to tasks commensurate with their task-relevant knowledge and skills. | I13.2 | |
| | | We ensure that there is compatibility between group members' expertise and work processes. | I13.3 | |
| | | Overall, our group is well coordinated. | I13.1 | |

**Table 1.** *Cont.*

| Authors | Base Literature | Measurement Items | GMRG Survey Item | Researched Industry |
|---|---|---|---|---|
| Wu et al. (2010) [8] | (Teece et al., 1997 [2]; Swink and Hegarty, 1998 [54]; Sen and Egelhoff, 2000 [55]; Schroeder et al., 2002 [37]; Subramaniam and Youndt, 2005 [56]) | Operational cooperation | | Theoretical paper |
| | | Our information system facilitates cooperation across functions. | I13.1-3. | |
| | | Our formal procedures facilitate teamwork across functions. | I13.5-8. | |
| | | Our employees are skilled at maintaining healthy relationships with each other to diagnose/solve problems. | I13.1. | |
| | | Our employees are skilled at partnering with suppliers/clients to develop solutions for improvement. | I13.13-18. | |
| | | Operational customization | | |
| | | Our equipment has been used in unique ways that differentiate us from our competitors. | CG09.6 | |
| | | Our product design process has been modified and extended to better serve the needs of our customers. | CG09.3 | |
| | | Our planning systems have been modified and extended to better serve the needs of our customers. | FC2.2 | |
| | | Our production process has been modified and extended to gain unique positions in the market. | FC2.1. | |
| | | We have introduced new, internally developed materials into our employee training programs. | FC8.1-3. | |
| | | We stimulate teamwork to facilitate the sharing of individual knowledge throughout the organization. | I13.1-4. | |
| | | Operational responsiveness | | |
| | | We reduce uncertainty of equipment availability by quickly and easily changing the route of a job flow. | CG09.3 | |
| | | We adjust for unexpected variations in components and material inputs easily and quickly. | CG09.3 | |
| | | We adjust for unexpected variations in labor requirements easily and quickly. | CG09.3 | |
| | | We adjust for the unexpected changes in shipment requirements easily and quickly. | CG09.3 | |
| | | Operational improvement | | |
| | | We continuously standardize production processes. | I13.5 | |
| | | We continuously simplify production processes. | FC4.3 | |
| | | We continuously reduce waste and variance. | I13.9-11. | |
| | | We have learned from past successes and failures to improve processes continuously. | I13.12 | |
| | | Operational innovation | | |
| | | We have created innovations that made our prevailing processes obsolete. | FC8.1 | |
| | | We have created innovations that fundamentally changed our prevailing processes. | FC8.2 | |
| | | We have created innovations that made our existing expertise in prevailing processes obsolete. | FC8.3 | |
| | | Operational reconfiguration | | |

**Table 1.** *Cont.*

| Authors | Base Literature | Measurement Items | GMRG Survey Item | Researched Industry |
|---|---|---|---|---|
| Kump, et al. (2019) [9]. | Mandal (2017) [57]; Pandit et al. (2017) [58]; Rashidirad, et al. (2017) [59]; Babelytė-Labanauskė and Nedzinskas (2017) [60]; Lopez-Cabrales, Bornay-Barrachina, and Diaz Fernandez (2017) [61]; Shafia et al. (2016) [62] | We sense/are aware of the change of the environment. | FC8.4 | Austria 2013, 307 companies |
| | | We adopted new and better practices to respond to market changes. | FC8.4 | |
| | | We reconfigure (combine/release) resources to respond to market changes. | I13. 9-12. | |
| | | We develop competence and skills to respond to market changes. | I13. 9-12. | |
| | | SE1 Our company knows the best practices in the market. 0.72 | I11a | |
| | | SE2 Our company is up to date on the current market situation. 0.82 | I11b | |
| | | SE3 Our company systematically searches for information on the current market situation. 0.95 | I11c | |
| | | SE4 As a company, we know how to access new information. 0.83 | I11d | |
| | | SE5 Our company always has an eye on our competitors' activities. 0.70 | I11e | |
| | | SE6 Our company quickly notices changes in the market. 0.40 0.48 | FC8.4 | |
| | | SZ1 Our company can quickly relate to new knowledge from the outside. 0.87 | I12a | |
| | | SZ2 We recognize what new information can be utilized in our company. 0.71 | I12b | |
| | | SZ3 Our company is capable of turning new technological knowledge into process and product innovation. 0.84 | I12c | |
| | | SZ4 Current information leads to the development of new products or services. 0.73 | I12d | |
| | | T1 By defining clear responsibilities, we successfully implement plans for changes in our company. 0.89 | CG09a | |
| | | T2 Even when unforeseen interruptions occur, change projects are seen through consistently in our company. 0.90 | CG09b | |
| | | T3 Decisions on planned changes are pursued consistently in our company. 0.61 | CG09c | |
| | | T4 In the past, we have demonstrated our strengths in implementing changes. 0.60 | Cg09d | |
| | | T5 In our company, change projects can be put into practice alongside the daily business. 0.72 | Cg09e | |
| | | T6 In our company, plans for change can be flexibly adapted to the current situation. 0.71 | CG09f | |

Table 1 shows prior measurement models up to now that have been proposed for measuring DCs. Secondly, it shows that all the variables are covered in the GMRG V measurement instrument (except benchmarking from [6]). We can conclude that the GMRG measurement instrument is applicable for our research on the influence of capabilities on operating and business performance. This is not a surprise, since the development of GMRG survey is heavily grounded in the literature and rigorous academic research. From the variables shown in the fourth column of Table 1, where GMRG variables are presented, the variables are from different sections of the GMRG instrument (operational practices, innovation, and factory culture). This fact diminishes the common method

variance problem. In practice, these questions are often answered by different people in respondent firms.

Inspecting Table 1, dynamic capabilities include organizational change, developing internal and external ties, and information sharing and learning.

### 4.2. Method

For the analysis, we used the GMRG V dataset. We provided three analyses: one performed with structural equation modeling (SEM), the other using the Heckman two-step procedure to check for selectivity bias, and finally, analysis using OLS regression. Validation of mediation and moderation was performed through both SEM and OLS regression to check for the consistency of results.

### 4.3. Sample

The data were obtained from Round V of the GMRG data collection effort. A description of the Global Manufacturing Research Group (GMRG) (www.gmrg.org, accessed on 11 June 2023) can be found in [63,64]. Existing constructs and measures were used to ensure their validity, and the standardized survey instrument was developed in English. The process of designing the survey is described in [65]. Translation and back translation are described in [66,67]. The unit of analysis was the manufacturing site or plant. The collection process is described in [29,42]. Prior to each round, detailed data package instructions were provided to all data gatherers. They did not have to gather a lot of data (the minimum was set to 30 plants), but those that that were collected had to be collected with utmost attention. Usually, several responders at manufacturing plants were contacted, because in the first part of the questionnaire under Demographics, 15 questions were related to revenues and cost, while others responded to questions on manufacturing characteristics, innovation, supply chain, sustainability, and culture. A chi-square test between the first 15 and last 15 answers on several variables was performed for each country, and there was no evidence of non-response bias. In the sample of 1008 manufacturing plants, 27.1% were small companies with up to 50 employees, 41.4% of companies were middle-sized companies (with 50 to 250 employees) and 31.5% of companies were large, with over 250 employees. The data were obtained from 16 countries and 21 industry classifications. The data possessed adequate variety for generalizability [66]. The distribution of the sample by country is shown in Table 2.

**Table 2.** The sample.

| | Frequency | Percent | Profit Margin | % Sales by New Products | R&D Budget | Process Technology Investment Budget | Training Budget |
|---|---|---|---|---|---|---|---|
| Australia | 74 | 7.3 | 21.71 | 17.55 | 3.58 | 3.08 | 2.73 |
| Canada | 4 | 0.4 | 24.5 | 13.75 | 3.22 | 3.22 | 2.78 |
| China | 102 | 10.1 | 12.49 | 36.22 | 3.47 | 3.31 | 2.93 |
| Croatia | 113 | 11.2 | 18.05 | 24.54 | 2.83 | 3.53 | 2.43 |
| Czech | 1 | 0.1 | | 20.00 | 3.22 | 3.22 | 2.78 |
| Germany | 45 | 4.5 | 16.33 | 28.88 | 3.22 | 3.22 | 2.78 |
| Hungary | 38 | 3.8 | 7.72 | 19.51 | 2.35 | 2.7 | 2.01 |
| India | 58 | 5.8 | 23.77 | 22.24 | 3.76 | 3.33 | 3.6 |
| Ireland | 30 | 3 | 25.05 | 23.66 | 3.37 | 2.33 | 3 |
| Netherlands | 2 | 0.2 | 24 | 12.50 | 3.22 | 3.22 | 2.78 |
| Nigeria | 50 | 5 | 13.06 | 24.86 | 3.22 | 3.22 | 2.78 |
| Poland | 80 | 7.9 | −4.14 | 25.50 | 2.14 | 2.69 | 2.25 |
| Taiwan | 80 | 7.9 | 18.28 | 14.92 | 3.22 | 3.1 | 2.65 |
| Ukraine | 50 | 5 | 21.1 | 20.35 | 3.22 | 3.22 | 2.78 |
| USA | 168 | 16.7 | 16.1 | 20.89 | 3.39 | 3.21 | 2.89 |
| Vietnam | 113 | 11.2 | 18.14 | 42.46 | 3.67 | 3.75 | 3.09 |
| Total | 1008 | 100 | | | 3.22 | 3.22 | 2.78 |
| Average | | | 14.99 | 25.50 | 0.51–0.75% sales | 5–8% sales | 1.1–1.5% sales |

Table 2 also shows average Profit Margins, Percentage of Sales by New Products, average R&D Budget, Process Technology Investment Budget, and Training Budget.

### 4.4. Control Variables

Size [68,69] was used as a standard control variable. The second control variable was industry according to the complexity of the product, as explained in [70]. We did not use SIC codes, as it is known that in one SIC code, there might be simple and complex production processes. An additional control variable was used for developed or developing countries [29,71].

### 4.5. Measures

Independent variables of operations capability and dynamic capabilities were derived from the literature. We report only the model fit for each construct, and then in Table 3 we present the whole model with all constructs. Operations performance is a second-order construct including different questions on quality, cost, delivery, flexibility, and innovation.

**Table 3.** Confirmatory factor analysis of the proposed model.

| Reliability | GMRG Survey Question | GMRG Code | Construct | Estimate | S.E. | C.R. | *p* |
|---|---|---|---|---|---|---|---|
| | sensing | <--- | dinacap | 0.48 | | | |
| | coordination | <--- | dinacap | 0.967 | 0.207 | 10.472 | *** |
| | transformation | <--- | dinacap | 0.372 | 0.143 | 8.017 | *** |
| | budget | <--- | dinacap | 0.144 | 0.127 | 3.38 | *** |
| | learning | <--- | dinacap | 0.871 | 0.188 | 9.817 | *** |
| | I13h | <--- | coordination | 0.717 | | | |
| | I13g | <--- | coordination | 0.645 | 0.031 | 30.45 | *** |
| | I13a | <--- | coordination | 0.388 | 0.051 | 10.473 | *** |
| | I13b | <--- | coordination | 0.597 | 0.049 | 16.328 | *** |
| | I13c | <--- | coordination | 0.652 | 0.046 | 18.026 | *** |
| | I13d | <--- | coordination | 0.642 | 0.049 | 18.142 | *** |
| | I13o | <--- | coordination | 0.751 | 0.047 | 19.405 | *** |
| | I13p | <--- | coordination | 0.725 | 0.048 | 19.268 | *** |
| | I13q | <--- | coordination | 0.645 | 0.05 | 17.947 | *** |
| | I13r | <--- | coordination | 0.704 | 0.045 | 19.678 | *** |
| | I13s | <--- | coordination | 0.712 | 0.046 | 19.797 | *** |
| | I13t | <--- | coordination | 0.683 | 0.049 | 19.215 | *** |
| | FC01fb | <--- | sensing | 0.591 | | | |
| | FC01ec | <--- | sensing | 0.514 | 0.059 | 15.301 | *** |
| CR = 0.734, AVE = 0.417, Alpha = 0.915 | FC01ea | <--- | sensing | 0.641 | 0.066 | 15.644 | *** |
| | FC01cd | <--- | sensing | 0.683 | 0.072 | 15.82 | *** |
| | FC01ca | <--- | sensing | 0.56 | 0.064 | 14.352 | *** |
| | FC01bb | <--- | sensing | 0.683 | 0.076 | 14.807 | *** |
| | FC01ab | <--- | sensing | 0.75 | 0.076 | 17.034 | *** |
| | FC01aa | <--- | sensing | 0.803 | 0.079 | 17.256 | *** |
| | I13f | <--- | learning | 0.623 | | | |
| | I13e | <--- | learning | 0.646 | 0.043 | 21.522 | *** |
| | I13n | <--- | learning | 0.72 | 0.059 | 16.281 | *** |
| | I13m | <--- | learning | 0.813 | 0.06 | 17.704 | *** |
| | I13l | <--- | learning | 0.793 | 0.063 | 17.458 | *** |
| | I13i | <--- | learning | 0.758 | 0.06 | 16.33 | *** |
| | I06e | <--- | transformation | 0.908 | | | |
| | I06d | <--- | transformation | 0.873 | 0.026 | 35.695 | *** |
| | I06c | <--- | transformation | 0.89 | 0.026 | 36.93 | *** |
| | I06b | <--- | transformation | 0.795 | 0.026 | 31.004 | *** |
| | I06a | <--- | transformation | 0.793 | 0.027 | 29.575 | *** |
| | R_Dbudget | <--- | abscap | 0.646 | | | |
| | proces_budget | <--- | abscap | 0.69 | 0.056 | 14.416 | *** |
| | training_budget | <--- | abscap | 0.733 | 0.065 | 14.472 | *** |

**Table 3.** *Cont.*

| Reliability | GMRG Survey Question | GMRG Code | Construct | Estimate | S.E. | C.R. | *p* |
|---|---|---|---|---|---|---|---|
| | CG09f | <--- | operatcapability | 0.601 | | | |
| | CG09e | <--- | operatcapability | 0.543 | 0.053 | 17.907 | *** |
| CR = 0.844, AVE = 0.478, Alpha = 0.851 | CG09d | <--- | operatcapability | 0.77 | 0.061 | 16.876 | *** |
| | CG09c | <--- | operatcapability | 0.712 | 0.058 | 15.876 | *** |
| | CG09b | <--- | operatcapability | 0.751 | 0.057 | 16.378 | *** |
| | CG09a | <--- | operatcapability | 0.739 | 0.061 | 16.249 | *** |
| CR = 0.885, AVE = 0.720, Alpha = 0.880 | CG11a | <--- | busPerf | 0.876 | | | |
| | CG11b | <--- | busPerf | 0.863 | 0.029 | 31.836 | *** |
| | CG11c | <--- | busPerf | 0.804 | 0.025 | 29.363 | *** |
| | cost | <--- | OpPerf | 0.546 | | | |
| | quality | <--- | OpPerf | 0.814 | 0.111 | 11.483 | *** |
| | delivery | <--- | OpPerf | 0.815 | 0.139 | 11.986 | *** |
| | flexibility | <--- | OpPerf | 0.864 | 0.142 | 12.455 | *** |
| | innovation | <--- | OpPerf | 0.555 | 0.117 | 10.947 | *** |
| | CG10a | <--- | cost | 0.729 | | | |
| | CG10b | <--- | cost | 0.865 | 0.058 | 20.992 | *** |
| | CG10c | <--- | cost | 0.66 | 0.044 | 18.532 | *** |
| | CG10d | <--- | quality | 0.702 | | | |
| CR = 0.848, AVE = 0.536, Alpha = 0.885 | CG10e | <--- | quality | 0.761 | 0.055 | 19.634 | *** |
| | CG10f | <--- | quality | 0.76 | 0.074 | 16.517 | *** |
| | CG10g | <--- | delivery | 0.805 | | | |
| | CG10h | <--- | delivery | 0.793 | 0.036 | 26.304 | *** |
| | CG10i | <--- | delivery | 0.879 | 0.047 | 22.554 | *** |
| | CG10j | <--- | flexibility | 0.849 | | | |
| | CG10k | <--- | flexibility | 0.799 | 0.04 | 24.156 | *** |
| | CG10l | <--- | innovation | 0.845 | | | |
| | CG10m | <--- | innovation | 0.801 | 0.052 | 18.476 | *** |
| | CG10n | <--- | innovation | 0.856 | 0.042 | 22.607 | *** |

Overall model fit: $\chi^2$ = 2.931 < 3, GFI = 0.863, NFI = 0.888, IFI = 0.917, CFI = 1.000, all close to 0.9. REMSA = 0.045, PCLOSE = 1.000. Legend: <--- displays the direction to components that form the construct. *** means significance is at level $p < 0.001$.

Operations capability is derived from the work of [4,6,8,72], containing five statements and measuring the agreement with statements on a 7-point Likert scale. The operations capability construct tested individually showed a good model fit: $\chi^2$ = 0.93 < 3, GFI = 0.999, NFI = 0.999, IFI = 1.000, and CFI = 1.000; all close to 0.9. REMSA = 0.000 < 0.05, PCLOSE = 0.936, and Cronbach's Alpha reliability of construct = 0.851.

Dynamic capability is a second-order construct because it includes sensing, seizing and reconfiguring. Table 1 provides the basis to construct the dynamic capability measurement instrument. Questions from the GMRG survey reported in Table 1 are grouped so that they mostly resemble Teece's [26] definition of sensing, seizing, and reconfiguring; however, we tried to use the naming from the authors in Table 1 for completeness, because these authors were the first to make attempts to measure dynamic capabilities. Therefore, we divided dynamic capabilities into sensing, absorptive capability, learning, coordination, and transformation. Even though the model fit was good, we reduced the model to include only variables with factor loadings over 0.6. The model fit for confirmatory factor analysis of the dynamic second-order construct was as follows: $\chi^2$ = 3.179 < 5, GFI = 0.932, NFI = 0.946, IFI = 0.962, and CFI = 0.962, all close to 0.9. REMSA = 0.047 < 0.05, PCLOSE = 0.917, and Alpha reliability of construct = 0.915.

Operations performance was measured using 14 statements measured against a firm's competition adapted from [42] and under the assumption that managers can accurately assess performance, which was proven in [73]. This is also a second-order construct. The model fit for confirmatory factor analysis of the operations performance construct was as

follows: $\chi^2$ = 5.697, GFI = 0.947, NFI = 0.945, IFI = 0.954, and CFI = 0.954; all close to 0.9. REMSA = 0.069, PCLOSE = 0.0, and Alpha = 0.885.

The second dependent variable was business performance, adapted from [6], which had to be computed by the financial department for the last two years. The responders had to assess the total sales, profitability and market share rise or fall in the last two years. Seven categories were provided: 1 (reduced more than 25%) to 7 (rose by more than 25%).

Capabilities and performance measures were on different pages to avoid common method bias in accordance with [74–76]. One-factor analysis accounted for only 26.6% of variance, which is less than 30%, and therefore acceptable [74]. Capabilities and performance were all multi-item, second-order constructs, and as such were more reliable ([77], p. 462).

First, confirmatory factor analysis was performed on constructs that we proposed, since they are derived from theory and thus we hypothesized factor membership. This was achieved using AMOS 29, because AMOS allows for forming factor membership—that is, confirmatory factor analysis—unlike SPSS, which allows for only exploratory factor analysis. The model fit for all constructs was satisfied and is reported in Table 3. All measures are in an appropriate range. The goodness of fit for all of our constructs was good, and the factor loadings were all above 0.6 (0.7 is prescribed by [78]); ref. [79] states that factor loadings of even 0.3 are acceptable as long as the sample comprises more than 350 cases, which is indeed valid for our sample of 1008 manufacturing plants. Table 3 presents the whole measurement model and is the basis for further replication.

Model fit was good, so there was no reason to abandon our analysis in AMOS with our theoretically constructed items. As can be seen in Table 3, all threshold values were all in acceptable ranges (the overall model fit very well: $\chi^2/\mathrm{df}$ = 2.931 < 3, GFI = 0.863, NFI = 0.888, IFI = 0.917, CFI = 1.000, all close or above 0.9, REMSA = 0.045 < 0.05, PCLOSE = 1.000) [80]. Composite reliability (CR) was satisfactory because all values were >0.7 [81]. The results also indicated acceptable discriminant validity for the measures at both the construct and item levels. The average variances extracted (AVEs) were greater than the squared correlation of the construct [81] and >0.5 [82]. All of our AVEs were close to or higher than 0.5.

Cronbach's Alpha was also computed (internal consistency) as displayed in Table 3, under CR and AVE, and values were >0.7, as prescribed by [83]. Inter-correlations of items are presented in Table 4.

**Table 4.** Interim correlations (** denotes significance at *p* < 0.01).

|  | Mean | Std. Deviation | N | busPerf | OpPerf | Operat Capability | Dinacap |
|---|---|---|---|---|---|---|---|
| busPerf | 4.50 | 1.19 | 975 | 0.848 |  |  |  |
| OpPerf | 2.90 | 0.44 | 975 | 0.301 ** | 0.732 |  |  |
| operatcapability | 4.75 | 0.99 | 975 | 0.326 ** | 0.652 ** | 0.691 |  |
| dinacap | 2.46 | 0.30 | 975 | 0.258 ** | 0.437 ** | 0.419 ** | 0.645 |

Measurement equivalence with GMRG IV was tested by [42,65], and measurement equivalence was established. The procedure consisted of clustering companies into three country development groups; developed, emerging and developing. For the test for this analysis, we used data from the World Economic Forum [84] and ranked countries according to their competitive index: countries with a competitive index over 5 were coded as developed, and the others were coded as developing. The USA, Germany, Netherlands, Canada, Australia, and Ireland were classified as developed countries, while the rest (China, Taiwan, Czech Republic, Poland, India, Vietnam, Hungary, Croatia, Ukraine, and Nigeria) were assigned to the developing group of countries. This procedure had to be performed in AMOS, as we had to make two groups for our model: one for developed and one for developing countries. Then, we ran a two-group confirmatory factor analysis and obtained good model fit ($\chi^2/\mathrm{df}$ = 2.607 < 3, GFI = 0.806, NFI = 0.816, IFI = 0.878, and CFI = 0.877;

all close to 0.9, REMSA = 0.040 < 0.05 and PCLOSE = 1.000), establishing that there is measurement equivalence among the GMRG V sample as well.

Common method bias was tested in accordance with [75,76]. All four recommendations [74,85] were included in the design phase of the questionnaire. In this work, all four preconditions were fulfilled.

Next, we tested for selectivity bias. Heckman is very strict about selectivity bias, because not all companies are surveyed. Therefore, we checked whether there was some bias in our sample, such as only including better-performing companies [86].

In order to check for bias, the Mills ratio was calculated [87]. In our case, for operation performance as a dependent variable, we obtained $R^2$ = 0.1973, Inverse Mills Ratio = 0.268 (sig. 0.054), and Rho = 1.0313, and for business performance, we obtained $R^2$ = 0.1382, Inverse Mills Ratio = 0.408 (sig. 0.366), and Rho = −0.9293. We observed a positive nonsignificant selection bias for both models, with operations performance and business performance as dependent variables, and operational and dynamic capability as independent variables.

*4.6. Results*

To test our research question on how operations and dynamic capabilities affect operations and business performance, we built the base model shown in Table 5 using AMOS with bootstrapping to capture direct and indirect effects.

**Table 5.** Results of Models 1 and 2.

| | Model 1 | | Model 2 | | |
|---|---|---|---|---|---|
| |  | |  | | |
| Direct effects | Operations performance | Business performance | Dynamic capability | Operations performance | Business performance |
| Operations capability | 0.598 (0.023) | 0.201 (0.007) | 0.419 (0.000) | 0.569 (0.000) | 0.197 (0.000) |
| Dynamic capability | 0.209 (0.008) | 0.126 (0.008) | | 0.199 (0.000) | 0.124 (0.000) |
| Operations performance | | 0.114 (0.004) | | | 0.118 (0.004) |
| Indirect effects | | | | | |
| Operations capability | 0.000 (-) | 0.081 (0.016) | 0.000 (-) | 0.000 (-) | 0.081 (0.016) |
| Dynamic capability | 0.000 (-) | 0.092 (0.010) | | 0.349 (0.023) | 0.529 (0.013) |
| Operations performance | | 0.000 (-) | | | 0.000 (-) |
| Model Fit | $\chi^2$ = 2.830 < 3, GFI = 0.867, NFI = 0.885, IFI = 0.922, CFI = 0.922, all close to 0.9. REMSA = 0.043 < 0.05, PCLOSE = 1.000 | | $\chi^2$ = 2.761 < 3, GFI = 0.869, NFI = 0.888, IFI = 0.925, CFI = 0.925, all close to 0.9. REMSA = 0.042 < 0.05, PCLOSE = 1.000 | | |

We can see that there is evidence of a strong direct relationship between both operating capabilities and dynamic capabilities and performance (Model 1). This proves the first hypothesis that there is a strong significant positive influence of both operating and dynamic capabilities on operating and business performance. Indirect effects are almost negligible. This is in line with theoretical prescriptions provided by [2,3] and proved the results from recent measurement propositions published by [9,10], but is in contradiction with the findings of [49,50], as we hypothesized.

In Model 2, when introducing the mediating effect of dynamic capabilities on operations capabilities, we observe a strong direct effect of dynamic capabilities on operating capabilities and a simultaneous reduction in the standardized regression weights of the effects of operations capabilities on operating and business performance, meaning that we can perceive a partial mediation (not full mediation because the significances did not become insignificant). In parallel, we also obtain the direct and indirect effects of dynamic capabilities on both operations and business performance. Interestingly, when mediation is included, we can see a strong significant indirect effect of dynamic capabilities on operating and business performance. This result is in line with [6].

Our results indicate the presence of a significant positive influence of capabilities on operations and business performance in our sample. This is not generalizable according to [88], but it is a starting point for further discussion and replication. According to [36], because of causal ambiguity—that is, because it takes time for an initiative to produce results—normative prescriptions are often inappropriate. Some operations capabilities can be very valuable sources of competitive advantage in some industries and at some points of time, but there is no generally valuable advice for all industries in all times, and there is no universal advice for firms in terms of in which capabilities to invest. According to [13], each company should perform a type of cost–benefit analysis regarding into which capabilities to invest first.

The next hypothesis sought to test whether there is a moderating effect. The analysis was performed in both AMOS and SPSS. We imputed our constructs created in AMOS into SPSS and then had to standardize values to calculate the moderating effect. We had to compute the moderating effect obtained as a multiplication of the standardized values of dynamic capability and operations capability. For this analysis, unstandardized regression values had to be entered into a graphing tool.

The AMOS text output is presented in Table 6.

**Table 6.** Results from AMOS analysis (*** denotes significance at $p < 0.001$ and <--- shows the direction of the relationship).

| | | | Estimate | S.E. | C.R. | $p$ |
|---|---|---|---|---|---|---|
| ZOpCap_X_ZDinCap | <--- | Zdinacap | −0.014 | 0.035 | −0.4 | 0.689 |
| ZOpCap_X_ZDinCap | <--- | Zoperatcapability | 0.086 | 0.035 | 2.433 | 0.015 |
| Zoperationsperformance | <--- | Zdinacap | 0.199 | 0.026 | 7.647 | *** |
| Zoperationsperformance | <--- | Zoperatcapability | 0.57 | 0.026 | 21.868 | *** |
| Zoperationsperformance | <--- | ZOpCap_X_ZDinCap | −0.009 | 0.024 | −0.362 | 0.717 |
| ZbusinessPerformance | <--- | Zdinacap | 0.124 | 0.034 | 3.659 | *** |
| ZbusinessPerformance | <--- | Zoperatcapability | 0.201 | 0.04 | 5.004 | *** |
| ZbusinessPerformance | <--- | Zoperationsperformance | 0.117 | 0.041 | 2.899 | 0.004 |
| ZbusinessPerformance | <--- | ZOpCap_X_ZDinCap | −0.041 | 0.03 | −1.372 | 0.17 |

We can see that the moderating effect is insignificant, and thus we have no evidence of a moderating effect of dynamic capabilities on operating and business performance.

From Table 6 and Figure 4, one can see that moderating values are nonsignificant in relation to operating and business performance, meaning that we observed no moderating effect. Therefore, the third hypothesis about the moderating role of dynamic capabilities must be rejected.

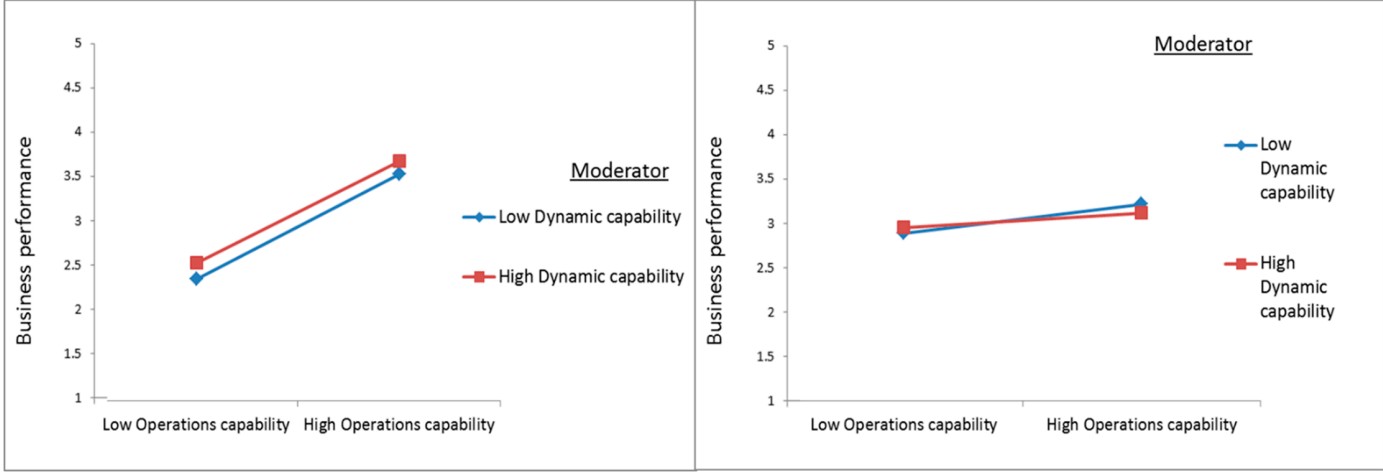

**Figure 4.** Moderation results.

In SPSS, we again calculated standardized values and performed two-stage OLS regressions for both of our depended variables, but this time we looked at $\Delta R^2$ change and whether the moderating effect was significant or not. The results are similar to in AMOS, and the interaction terms have nonsignificant effects. However, operations performance directly affects business performance, in line with the AMOS results—that is, operations performance (B = 0.139, *p* = 0.001) significantly and positively affects business performance. This is in line with [89], who proposes that competitive advantage leads to superior business performance. Interestingly, none of the control variables matter except for the size of the company. Smaller companies obtain better operations performance, while larger companies obtain better business results, as can be seen from Table 6 containing the results of OLS regression in SPSS.

If we look at Table 7, we can see that size is significant in operating performance and business performance, because significances are *p* < 0.001. However, operating performance slightly favors smaller companies, but in case of business performance bigger companies obtain better business performance. Again, it can be seen as in Table 6, that operations capability and dynamic capability significantly and positively affect operating (first three columns, rows 5 and 6) and also operations capability, dynamic capability and operations performance significantly and positively affect business results (columns 4–6, rows 5–7).

**Table 7.** SPSS OLS regressions with operations and business performance as dependent variables.

| | **Operations Performance** | | | **Business Performance** | |
| --- | --- | --- | --- | --- | --- |
| | **Standard. Beta** | **Sig.** | | **Standard. Beta** | **Sig.** |
| (Constant) | | <0.001 | (Constant) | | 0.002 |
| noofemployee | −0.086 | <0.001 | noofemployee | 0.152 | <0.001 |
| Complexity | −0.03 | 0.222 | Complexity | 0.034 | 0.266 |
| Developed or developing | −0.045 | 0.068 | Developed or developing | 0.019 | 0.539 |
| Zscore(operatcapability) | 0.575 | <0.001 | Zscore(operatcapability) | 0.168 | <0.001 |
| Zscore(dinacap) | 0.2 | <0.001 | Zscore(dinacap) | 0.118 | <0.001 |
| | | | Zscore(operationsperformance) | 0.139 | <0.001 |
| ZOpCap_X_ZDinCap | −0.001 | 0.953 | ZOpCap_X_ZDinCap | −0.044 | 0.147 |

## 5. Discussion of Results

In this work, we analyzed three rival models: Model 1—direct model; Model 2—mediation model; and Model 3—moderation model. The best model fit was obtained by Model 2, the mediation model, even though only partial mediation was established. This is further reinforced by the fact that dynamic capabilities, even in this mediating model, have

significant indirect effects on operations and business performance, but strong direct effects on operating capabilities. Operations capabilities by far have the most significant impact on operating performance. In the direct model—Model 1—operations capabilities have stronger effects on business performance, but when mediation is introduced—Model 2—dynamic capabilities start to contribute more to business performance, both directly and indirectly. In Model 3, the moderation of dynamic capabilities does show good model fit but the moderating variable is insignificant, enabling us to state that no moderation effects are present. Again, we find strong direct effects of dynamic capabilities on operating capabilities and strong indirect effects of dynamic capabilities on operating and business performance. We have to conclude that dynamic capabilities directly affect operations capabilities and, in this way, enhance business and operating performance both directly and indirectly. Model 1 provides confirmation of [2,3,46–48], which also claim that dynamic capabilities enhance performance.

In Model 2, dynamic capabilities start to contribute more to business performance, both directly and indirectly, somewhat in line with Pavlou and El Sawy [7], and contradicting [47,48], who found no mediating relationship. H2 is in line with [26,35], who also propose the mediating relationship. In Model 3, the moderation of dynamic capabilities does show good model fit, but the moderating variable is insignificant, enabling us to state that no moderation effects are present. This is not in line with line with [31,48], who claim that organizational competencies are the moderating variable. Again, we find strong direct effects of dynamic capabilities on operating capabilities and strong indirect effects of dynamic capabilities on operating and business performance. We have to conclude that dynamic capabilities directly affect operations capabilities and, in this way, enhance business and operating performance both directly and indirectly.

All this means that if management invests into dynamic capabilities, it will bring about better operating and financial performance, thus making the company more competitive. The mediating effect is stronger, meaning that dynamic capabilities do not impact operating capabilities directly, and thus the statement from [15], who call the relationship "blurry", still stands.

Operating capabilities have by far the largest impact on operating performance. In order to build operations capabilities, the company should develop state-of-the-art manufacturing processes and superior knowledge and technological skills among its employees. This is in line with how Teece [26,28] defines operating capabilities, and is explained in [29]. Operating capabilities are affected by dynamic capabilities. It may be the relationship is quadratic, as explained by [20], finding that the relationship of intellectual capital with innovation is quadratic. In fact, we also performed linear, quadratic and cubic interpolation of dynamic capability and operations capability, but found no matching relationship even though curve estimations were significant and had an $R^2$ ranging from 0.12 to 0.18.

Therefore, the presence of dynamic capabilities in a manufacturing company's processes and routines creates a potential source of competitive advantage [90–93]. Dynamic capabilities necessitate learning, albeit different kinds of learning, depending on whether the intended outcome is innovation (will need more experimentation and problem solving) or knowledge articulation and codification [94], which is necessary for building dynamic capabilities. In fact, ref. [94] even state that sometimes it is better for a company to invest in the simple education of employees than to build dynamic capabilities. Dynamic capabilities involve, according to [94], learning. Learning is a social process that requires communication among employees, and through their communication, knowledge is articulated, becoming new knowledge that can then be codified. Foss et al. [94] explain this process very well. According to [26], developing dynamic capabilities resides in fast responses to perform everyday tasks by all employees. This capability of fast reconfiguration is not easily replicated. However, sensing, seizing, and reconfiguring are not always performed by top management, but also by employees. Employees at all levels sometimes sense and seize opportunities through constantly looking outside the company, adopting and changing procedures. Learning plays an important role, especially if new knowledge

comes from outside. Although internal knowledge is valuable, the assimilation of new knowledge is even better [5,95]. All of this is encapsulated in our proposed measurement model under the dynamic capability construct consisting of 30 statements, grouped into sensing, absorptive capability, learning, coordination, and transformation.

Therefore, dynamic capabilities contribute to operations performance directly and indirectly. Operations capabilities (ordinary capabilities) have the highest effect on operating and business performance. Operations capabilities, by aid of dynamic capabilities, lead to better operations performance, which directly leads to better business performance. This further supports the insights provided by [89].

In conclusion, our findings indicate that there is a significant relationship between operating capabilities and performance, but also between dynamic capabilities and performance, which is something that is not easily replicated. Building dynamic capabilities is a long-term process which consists of investing in employees, their communication, their learning abilities, and their sensing, seizing and reconfiguring abilities. Both contribute to operations performance (directly and indirectly), which directly relates to business performance.

## 6. Implications for Research and Practice

### 6.1. Theoretical Implications

The theoretical implications are manifested in this tested measurement instrument, which could be used by academics in various fields. Also, we tested for mediation and moderation and found mediation but not moderation. For academics, this is an opportunity to test the instrument in various settings and check the relationships.

This study provides a detailed and rigorous analysis of the largest global dataset of manufacturing practices in existence, involving much-used concepts such as dynamic and operating capabilities, and aspects of performance. These topics are of extreme practical importance to those investing in and managing such enterprises, and therefore to those who research such foundational concepts. We would argue that when such relatively complex concepts are in play, which are often not well defined in the literature, it is imperative that more studies such as ours are conducted to increase the clarity and empirical evidence base regarding these concepts and their interrelationships. There are clear overlaps between some of these concepts in practice, as shown in our research, which found partial support for mediation and no moderating relationships. Also, there is the problem that an operations capability in one industry or country might be thought of, in practical terms, as dynamic in another. However, we found some important relationships averaged across industries and countries in many stages of development that were pervasive.

Our findings generally support the concept and value of notions associated with dynamic capabilities, affecting operational and then business performance. While operating capabilities that lead to more foundational cost and quality outcomes were shown to be particularly important, these can be complemented by dynamic capabilities. We found only partial mediation and small, insignificant moderating effects, suggesting that the relationships are complex, which is nothing strange when human nature is involved. This should give confidence to practitioners who are looking to improve their intangible capital and their capabilities, both operational and dynamic.

While our study has a larger dataset and goes further than previous studies in trying to holistically relate some very important workplace concepts, there is much need to even further disentangle the core elements of which practices and capabilities work best. The better we can define these concepts and their relationships, the more clearly professional managers will be able to recognize, measure, and assess them in their workplaces, in order to implement changes and improvements. We propose that case studies should follow closely behind this study, aiming to uncover causal mechanisms and answer precisely how and why capabilities and dynamic capabilities affect performance. Such case studies will be able to triangulate with our results.

*6.2. Practical Implications*

Building dynamic capabilities is a long-term process which consists of investing in employees, their communication skills, their learning abilities, and their sensing, seizing and reconfiguring abilities. Both contribute to operations performance (directly and indirectly), which directly relates to business performance. According to [13], each company should perform a type of cost–benefit analysis into which capabilities to invest first. Sometimes it is more beneficial to invest in employees' skills and knowledge than to try to develop dynamic capabilities. However, if management decides to build dynamic capabilities, it is now proven that it would produce higher operating and financial performance and thus make the company more competitive, as the fathers of dynamic capabilities theory suggest.

*6.3. Limitation of the Study*

One limitation of the study was that it was conducted only on manufacturing companies, and we highly recommend future research on the application of this measurement instrument in other fields, such as service operations management.

## 7. Conclusions

This study contributes to the literature in various ways. First, we investigate the impact of capabilities on operational and business performance. This is a new contribution in the literature, as most researchers use either financial or operational measures for assessing performance. Secondly, derived from [25]'s suggestion to unbundle constructs, we were able to uncover interesting relationships that add to the body of knowledge on this topic.

In Table 3, we propose a comprehensive measurement model drawn from a literature search to measure capabilities and performance. Further refinement of the instrument should be performed, but we kept all the variables in the model to show the complexity of the problem of competitiveness and dynamic capabilities. In Table 1, we show that there is only limited research on how to measure dynamic capability, but our analysis in Table 1 shows that the GMRG research instrument can be effectively used for this complex task.

We clearly show that capabilities affect competitiveness measured in terms of increased operations performance, thus increasing business performance. It was possible to answer our questions using the large GMRG dataset comprising 1008 companies, and we applied multiple methods of analysis, and obtained good model fits. We can state on the grounds of our results that both capabilities (operating and dynamic) positively affect performance (operational and business performance). It was also seen that operations capability has the highest effect on operations performance, meaning that although dynamic capabilities do reinforce operations capabilities, ordinary capabilities are the base, and managers should not neglect operations capabilities. This is in line with the conclusions of [96] that operations capabilities change with time, and this change in operating capabilities becomes a dynamic capability.

Control variables showed that the size of the company matters: smaller companies have a better chance of profiting from dynamic capabilities, which is understandable because changing the culture is always easier in smaller environments than in larger companies.

We used data from the World Economic Forum [84] and ranked countries according to their competitive index: USA, Germany, Netherlands, Canada, Australia, and Ireland were classified as developed countries, while the rest (China, Taiwan, Czech Republic, Poland, India, Vietnam, Hungary, Croatia, Ukraine, and Nigeria) were assigned to the developing group of countries. This establishes that there is measurement equivalence among the GMRG V sample.

**Funding:** This research received no external funding.

**Informed Consent Statement:** Not applicable.

**Data Availability Statement:** The data are owned by Arizona State University who now lead the GMRG group. According to internal policy of the GMRG group, data are available only for data gatherers, since they invested a lot of work in collecting the data and guaranteeing anonymity to responding companies.

**Conflicts of Interest:** The author declares no conflict of interest.

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
