# Peer review of "Operating and Dynamic Capabilities and Their Impact on Operating and Business Performance"

_sustainability, doi:10.3390/su152015181_

Round 1
Reviewer 1 Report
Please, see the attached file.

The quality of English in the paper is average with some typos and writing mistakes.
Reviewer 2 Report
Thank you for the opportunity to review this study on organizational capabilities. First of all I'd like to ask: Why should this paper be considered for publication in Sustainability -journal? What is the contribution to the domain of sustainability?
However, my main concern with the paper is the use of concepts ordinary, operational and operative capabilities as well as the conceptualization of very generic dynamic capabilities.
After reading the paper I'm not convinced that the author has grasped the capabilities literature. At least it is not evident in the paper that uses the aforementioned concepts interchangeably without justifying the choice. The focus and the approach of the study is apparently on operations management literature, but it also tries to build on the strategic management stream without fully acknowledging that there are apparent differences.
The concept of ordinary capabilities is well established in the literature and there is wide agreement that those are the current capabilities that make it possible to for the organization to perform its current business. Sometimes also the term operational capabilities is used for this. However, operative capabilities as defined in the paper are a more narrow area of ordinary capabilities. So, looking at the way operational capabilities are defined in the paper, I interpret that operational capabilities are indeed ordinary capabilities, but there are also many other ordinary capabilities and these cannot be used as synonyms. (cf. Nand et al, 2023: DOI10.1109/TEM.2022.3152149)
Moreover, the study appears to assume that dynamic capability is a generic concept that can be measured without contextualization. This represents one of the two traditions within the dynamic capabilities research. However, also this choice remains unexplained in the study.
Therefore, I would ask the author to pay attention to the definitions of the key concepts and especially the conceptual relations between the many capability concepts included.
Because of these major conceptual concerns, I feel it's not meaningful to comment on the findings part at this stage.
Reviewer 3 Report
Dear Authors:
Entitled “Ordinary and Dynamic Capabilities and their impact on Operating and Business Performance” is a research study. The study proposes a new measurement model for measuring dynamic capabilities based on current propositions in literature. It also investigated the relevance of dynamic capabilities on firm competitiveness in manufacturing settings via research on how operating and dynamic capabilities affect operating and business performance. Additionally, this study showed that capabilities affect competitiveness measured in terms of increased operations performance, consequently increasing business performance. Both ordinary and dynamic capabilities positively affect operational and business performance. Therefore, ordinary capabilities are the base for obtaining competitive advantage.
Reviewer 4 Report
Thanks for the opportunity to review this article. The paper is well written, well structured, topical, brings an interesting contribution to the literature the author proposes a new model for measuring dynamic capabilities based on relevant aspects presented in the literature and quantifies their impact on operational and business performance, or more precisely the relevance of dynamic capabilities on the competitiveness of firms in the production environment, by investigating how operational and dynamic capabilities affect operational and business performance.
In the introduction the author reinforces the value of the proposed model, formulates the research questions and presents the structure of the research.
In the literature chapter the author reinforces that the presentation of ordinary and dynamic capabilities is focused more on innovation, but the role of capabilities on performance, can be extended in more sectors and types of economy. In this chapter, the author carries out a critical review of the literature on attempts to describe ordinary and dynamic capabilities by various authors, which even though they are presented quite often, models for measuring them have not been developed. Ordinary and dynamic capabilities should have a positive effect on operational and business performance, the author presents in this chapter 3 models testing the effects of dynamic capabilities on ordinary capabilities, the author proposes a number of 3 research hypotheses.
The research instrument is an appropriate one, for the analysis the author used the GMRG (Global Manufacturing Research Group V) dataset. The analysis carried out had 3 components: one carried out with structural equation modelling (SEM), the other using the two-stage Heckman procedure to check for selectivity bias and finally the OLS regression results.
The proposed measurement model was verified by structural equation modelling in SPSS AMOS software on data collected in GMRG V, self-reported using a database of 1008 production units from 16 countries. All tests performed both through SEM, SPSS AMOS and OLS in SPSS, as well as the two-stage Heckman procedure, produced consistent results and demonstrated a good fit of the proposed measurement model, which means that it can be used for further exploration of the interaction between regular and dynamic capacities.
Author Response
Thank you for reading the manuscript and your kind words.
Reviewer 5 Report
The paper is quite well written but the author in order to prove the feasibility of the model take into consideration companies operating in various fields of activity which have different dynamic and ordinary capabilities.
Considering the fact that and I quote: "Most of the research was conducted in high tech sector" perhaps the whole analysis would have been more appropriate to be carried out on this sector, not to mention the fact that in chapter 5. Discussion of results the focus is on dynamic capabilities in a manufacturing company’s processes.
Also bibliographic references are somehow out of date.
Reviewer 6 Report
Recommendations / suggestions:
- improve the manuscript abstract;
- more clearly emphasize the research objectives;
- argue more clearly the need for this research (based on literature);
- develop a general methodological scheme associated with the research [synthesis];
- highlight the contributions made through this research [theoretical and practical];
- improve the conclusions of the manuscript;
- improve the references section [update and develop];
- respect the requirements associated with the journal template.
Minor editing of English language required
Round 2
Reviewer 2 Report
I'm sorry my initial review came across as only negative. I had major concerns in terms of the core concepts of the study and wanted to highlight those so that the author could look into the issues. The revised version shows better consistency and clarity and thus the reader is better able to follow the logic and argumentation. The use of concepts is indeed very varied in the literature as the author points out in the paper and in the response. This is exactly why it is so critical to be very clear and consistent in current and future studies.
When it comes to the contribution of the study, the role of manufacturing operations in achieving improved sustainability should not be left between the lines, but it needs to be explicitly stated.
What comes to Teece (2007) article, the term sustainability refers in that article to sustaining operations and competitive advantage over time. There the focus is on the economic sustainability instead of sustainability in the triple bottom-line sense, which I have understood is this journal's approach to sustainability.
I recognize the work the author has put into revising the paper. Yet, I still find the framing of the study somewhat problematic. The way operational and dynamic capabilities are conceptualized and measured focuses on the manufacturing operations context. Therefore, I don't fully agree with the study being about overall dynamic capabilities. It seems that this study is about manufacturing related dynamic capabilities and the improvement of manufacturing capabilities. The measurement of dynamic capability includes many items/variables that are specifically about manufacturing operations. The contextualization is actually well brought up in the discussion section, where the author interprets the meaning of the findings in the manufacturing operations (for example in this statement: "the company should have state-of-the-art manufacturing processes and superior know how and technological skills of its employees"). However, this aspect is not very well visible in other parts of the study. Perhaps this limitation of the study could be at least mentioned in the paper to guide future research on the application of the measurement instrument.
All in all, I understand and acknowledge that there definitely is value added on constructing a measurement instrument for dynamic capability in the manufacturing context and examining the connection between operational and dynamic capabilities. In particular the focus on the importance of learning in the discussion is interesting and important.
Minor notion: on page 20 right below figure 4 the text refers to figure 2. Maybe the text should refer to fig. 4
